# Cloudy Apple Juice Fermented by *Lactobacillus* Prevents Obesity via Modulating Gut Microbiota and Protecting Intestinal Tract Health

**DOI:** 10.3390/nu13030971

**Published:** 2021-03-17

**Authors:** Mengzhen Han, Meina Zhang, Xiaowei Wang, Xue Bai, Tianli Yue, Zhenpeng Gao

**Affiliations:** College of Food Science and Engineering, Northwest A&F University, Yangling 712100, China; han_mengzhen@163.com (M.H.); meila_xile@163.com (M.Z.); wxw648454678@163.com (X.W.); snow205129@163.com (X.B.); yuetl305@nwsuaf.edu.cn (T.Y.)

**Keywords:** polyphenols, *Lactobacillus*, gut microbiota, intestinal tract health

## Abstract

Obesity and hyperglycemia are two serious chronic diseases that are increasing in incidence worldwide. This research aimed to develop a fermented cloudy apple juice with good hyperglycemia intervention activities. Here, cloudy apple juice (CAJ), cloudy apple juice rich in polyphenols (CAJP) and fermented cloudy apple juice rich in polyphenols (FCAJP) were prepared sequentially, and then the effects of the three apple juices on weight, lipid level, gut microbiota composition and intestinal tract health were evaluated for obese mice induced by a high-fat diet. The research findings revealed that the FCAJP showed potential to inhibit the weight gain of mice, reduce fat accumulation, and regulate the blood lipid levels of obese mice by decreasing the ratio of the Firmicutes/Bacteroidotas, improving the Sobs, Ace, and Chao indexes of the gut microbiota and protecting intestinal tract health. In addition, the FCAJP augmented the abundance of Akkermansia and Bacteroides, which were positively related to SCFAs in cecal contents. This study inferred that FCAJP could be developed as a healthy food for preventing obesity and hyperglycemia.

## 1. Introduction

Obesity is an important pathogenic factor for many chronic diseases such as diabetes, hypertension and coronary heart disease, and has become a serious public health issue around the world [1]. Obesity prevention is therefore an important research topic. However, medical treatments for obesity have been hampered by high costs and side effects [2]. The most effective way to prevent obesity is to restrict food and excessive energy intake, and increase energy consumption, for example, through exercise; however, few individuals can maintain these approaches over a long period of time [3]. A wide range of evidence has indicated that obesity is related to gut microbiota dysbiosis which in turn affects nutrition metabolism and energy absorption in body [1,4]. Gut microbiota participate in various metabolic activities in the human body that are closely related to health, such as regulating peristalsis and participating in the absorption of nutrients [5]. Under normal circumstances, the gut microbiota and the host will maintain a dynamic balance. If the dynamic balance is broken over a long time, it will cause disorders of the gut microbiota and lead to various diseases. A high fat diet can alter the composition of gut microbiota, leading to the occurrence of obesity and other metabolic disorders [6]. Therefore, regulating the balance of the gut microbiota by eating healthy food can prevent various diseases caused by gut microbiota without side effects [7].

Apple juice is rich in all kinds of polyphenols, which are ubiquitous secondary metabolites in plants. Evidence from both in vitro and in vivo studies has demonstrated that polyphenols have various biological effects, such as anti-inflammatory, antitumor, hepatoprotection and enhanced muscle contractile endurance [8], and regulation of gut microbiota [9]. Hence, cloudy apple juice (CAJ) is a potential functional product with hyperlipidemia intervention activity, and there is a hypothesis that the addition of polyphenols recovered from apple pomace to cloudy apple juice could not only enhance these healthy properties but also add value to the apple agroindustry. *Lactobacillus* has been reported to reduce weight gain via modulation of gut microbiota and metabolites [10,11]. Fruits and vegetables fermented by *Lactobacillus* not only exhibit increased nutrient content but also produce products that contain live bacteria and have a unique flavor [12]. Studies have shown that the total polyphenol content increased after fermentation by *Lactobacillus* [13]. Therefore, cloudy apple juice (CAJ), cloudy apple juice with the addition of polyphenols (CAJP) and fermented cloudy apple juice with the addition of polyphenols (FCAJP) were prepared sequentially to develop a functional cloudy apple juice.

There is limited research on the hyperlipidemia intervention activity of CAJ, CAJP and FCAJP on obese mice caused by a high-fat diet. Meanwhile, it is necessary to explore the hyperlipidemia intervention mechanism of these three kinds of apple juice. Therefore, in the present study, (1) cloudy apple juice (CAJ), cloudy apple juice rich in polyphenols (CAJP), and fermented cloudy apple juice rich in polyphenols (FCAJP) were prepared sequentially, then the hyperlipidemia intervention activity of these three apple juices on obese mice induced by a high-fat diet was evaluated; (2) the gut microbiota in fecal and short-chain fatty acids (SCFAs) in the cecum contents of mice was evaluated; (3) the intestinal inflammation, intestinal permeability were evaluated. The study will provide a theoretical basis for developing a functional apple juice and elucidating its hyperlipidemia intervention mechanism.

## 2. Materials and Methods

### 2.1. Microorganisms and Apples

Fresh Aksu Fuji (*Malus domestica* Borkh., Rosaceae), Qin Guan (*Malus domestica* Borkh., Rosaceae), and Granny Smith apples (*Malus domestica* Borkh., Rosaceae) were purchased from a local market (Yangling, China). *Lactobacillus acidophilus* 6005, *Lactobacillus plants* 21,805, and *Lactobacillus fermentum* 21,828 were all purchased from the China Center of Industrial Culture Collection (Beijing, China).

### 2.2. Methods

#### Preparation of Three Cloudy Apple Juices

A blended cloudy apple juice (CAJ) was prepared by mixing Aksu cloudy apple juice, Qin Guan cloudy apple juice and Granny Smith cloudy apple juice (volume ratio 4:3:1). Then, polyphenols were determined using HPLC (Alliance HPLC; Waters, -Hangzhou, -China) (chlorogenic acid: 143.989 ± 3.864 mg/L; gallic acid: 1.020 ± 0.051 mg/L; ellagic acid: 2.752 ± 0.659 mg/L; rutin: 0.126 ± 0.012 mg/L; phlorizin: 0.085 ± 0.001 mg/L; isoferulic acid: 0.160 ± 0.070; phloretin: 0.030 ± 0.006 mg/L).

Polyphenol-concentrated solutions were extracted from the apple pomace of Aksu, Qin Guan, and Granny Smith respectively using ultrasonic-assisted extraction (60% edible ethanol, 1:20 *w*/*v*, 45 °C, 50 min). Next the polyphenol-concentrated solutions were mixed in a volume ratio of 4:3:1. Then the blend polyphenol-concentrated solution was backfilled into the CAJ at 20% (*v*/*v*), to prepare cloudy apple juice rich in polyphenols (CAJP). Then, polyphenols were determined using HPLC (Alliance HPLC; Waters, Hangzhou, China) (chlorogenic acid: 132.222 ± 1.938 mg/L; gallic acid: 1.523 ± 0.335 mg/L; ellagic acid: 2.492 ± 0.074 mg/L; rutin: 2.963 ± 0.225 mg/L; phlorizin: 0.537 ± 0.002 mg/L; isoferulic acid: 1.531 ± 0.089; phloretin:0.033 ± 0.005 mg/L).

In order to develop the a fermented cloudy apple juice rich in polyphenols (FCAJP), *Lactobacillus acidophilus* 6005, *Lactobacillus plantarum* 21,805, and *Lactobacillus fermentum* 21,828 were cultured in MRS broth (Beijing Lugiao Co., Ltd., Beijing, China) twice prior to use (1% *v*/*v*, 37 °C, 18 h), which was more than 10^9^ CFU/mL. Subsequently, the CAJP was fermented with 3% (*v*/*v*) mixed *Lactobacillus* (volume ratio 1:1:1) at 37 °C for 24 h. The total number of viable cells was determined using standard decimal dilution method, which was 3.5 × 107 CFU/mL in FCAJP. Then, polyphenols were determined using HPLC (Alliance HPLC; Waters, Hangzhou, China) (chlorogenic acid: 138.239 ± 3.595 mg/L; gallic acid: 4.447 ± 0.204 mg/L; ellagic acid: 2.838 ± 0.103 mg/L; rutin: 1.572 ± 0.399 mg/L; phlorizin: 0.079 ± 0.003 mg/L; isoferulic acid: 1.997 ± 0.092; phloretin: 0.035 ± 0.001 mg/L). Finally, the three apple juices were stored at −30 °C within 3 days for administration to animals.

### 2.3. Animal Experiment Design

#### 2.3.1. Animals and Diet

Six-month-old male C57BL/6J mice (weighing 22–24 g) were purchased from the Experimental Animal Center of Xi’an Jiaotong University (SCXK [Shan] 2017–003). All experimental procedures are carried out in accordance with the “*Guidelines for the Care and Use of Laboratory Animals: Eighth Edition*” (ISBN-10:0-309-15396-4), and the experimental protocol has been approved by Northwest Agriculture and Forestry University. For 1 week, all mice could use water and standard diet freely. Then, the mice were randomly assigned to one of five groups (*n* = 12): the CD group (basal diet, saline); the HFD group (45% high-fat diet, saline); the HFD + CAJ group (45% high-fat diet, CAJ); the HFD + CAJP group (45% high-fat diet, CAJP); and the HFD + FCAJP group (45% high-fat diet, FCAJP (3.5 × 10^7^ CFU/mL)). Apple juices were administrated by oral gavage every day based on 0.15 mL/10 g body weight of mice (according to the equivalent dose conversion coefficient method, 0.15 mL/10 g body weight of mice is proximately equal to 140 mL/70 Kg of an adult) [14] and saline was used as control in CD and HFD groups. The basal diet (LAD3001M) and 45% high-fat diet (TP23100) were purchased from Trophic Animal Feed High-Tech (Nantong, China) [15]. The composition of basal diet and 45% high-fat diet was showed in (Appendix A). Daily intake of polyphenol of a mouse in five groups was presented in (Appendix A). The room was controlled to have a 12 h light/dark cycle and a constant temperature (22 ± 1 °C) and a relative humidity of 60−65%, with free access to water and basal diet/high-fat diet. After 8 weeks of dietary intervention, the mice were fasted for more than 12 h and anesthetized with 3.5% chloral hydrate [6]. Next, blood samples were collected from the orbital plexus and the organs were dissected and weighed. Liver index = liver weight (g)/body weight (g). The tissues that need to be immunohistochemically stored were preserved in 10% paraformaldehyde, and the remaining samples were stored at −80 °C for further use [3].

#### 2.3.2. Serum and Hepatic Parameter Analyses

Total cholesterol (TC), triglyceride (TG), low-density lipoprotein cholesterol (LDL-C), and high-density lipoprotein cholesterol (HDL-C) in serum (*n* = 8) were measured using an automatic biochemical analyzer (Chemray 800, Shenzhen Leidu Life Technology, Shenzhen, China) [11].

To prepare a 10% tissue homogenate, 0.5 g of liver tissue was added to 4.5 mL cold saline in a homogenate tube and broken with a crusher at 10,000 rpm. Then, the tissue homogenate was centrifuged at 2000 rpm for 15 min in a cryogenic centrifuge to obtain the supernatant for use [16].

The glutathione peroxidase (GSH-Px) superoxide dismutase, malondialdehyde (MDA) and protein content of liver (*n* = 8) were evaluated in accordance with the requirements of the kit (Nanjing Jiancheng Biological Technology Institute, Nanjing, China) [17].

#### 2.3.3. Histopathological Analysis

Approximately 50 mg epididymal white adipose tissues (EWAT), 8 mm colon tissue, and liver tissue from the same part of the mice were rinsed with saline and then fixed in 10% formaldehyde fixative for 18–24 h (*n* = 4). Afterwards, stained slides were made through standard procedures including dehydration, embedding, and hematoxylin staining. Finally, the stained slide was observed using an optical microscope (Olympus, Tokyo, Japan) [11]

#### 2.3.4. Sequencing of Fecal Microbiota

At the end of 8-weeks of experiments, a total of 40 fresh fecal samples with an average of eight in each group were randomly selected for sequencing. Briefly, DNA was extracted from samples using the DNA Kit (Omega Bio-tek, Norcross, GA, USA) according to manufacturer’s instructions and amplified for the V3 and V4 variable region of 16S rRNA genes using primer pairs: 338F (5′-ACTCCTACGGGAGGCAGCAG-3′) and 806R (5′-GGACTACHVGGGT-WTCTAAT-3′) with dual-index barcodes to tag each sample. Forty samples of PCR products were examined on an Illumina MiSeq platform by Shanghai Majorbio Bio-pharm Technology Co., Ltd. (Shanghai, China) [10].

The raw reads were deposited into the NCBI Sequence Read Archive (SRA) database. The raw 16S rRNA gene sequencing reads were demultiplexed, quality-filtered and merged. Operational taxonomic units (OTUs) with 97% similarity cutoff were clustered, and chimeric sequences were identified and removed. The data were analyzed on the free online platform of Majorbio I-Sanger Cloud Platform (www.i-sanger.com; accessed on 1 September 2020).

#### 2.3.5. Quantification of Short-Chain Fatty Acids (SCFAs) in Cecal Contents

SCFAs were extracted from samples as previously described by Xia [18]. About 100 mg of the cecal contents were dissolved in distilled water at a ratio of 1:10 (*w*/*v*); then the mixture was vortexed. The tube was allowed to stand in an ice water bath for 20 min and then centrifuged at 4800× *g* for 20 min at 4 °C. The 950 μL supernatant and 50 μL formic acid were mixed and filtered through a 0.22-μm nylon syringe filter (Waters). Subsequently, SCFAs were determined by gas chromatography (Agilent Technologies, Santa Clara, CA, USA) with a DB-FFAP capillary column (Agilent Technologies, Wilmington, DE, USA).

#### 2.3.6. Evaluation of Intestinal Permeability and Intestinal Inflammation

The levels of lipopolysaccharide (LPS), lipopolysaccharide binding protein (LBP) in serum, secretory immunoglobulin A (sIgA) and fecal calprotectin (FC) in fecal matter were determined using commercial mouse ELISA kits (Nanjing Jiancheng Biological Technology Institute).

### 2.4. Statistical Analysis

Data were analyzed using the SPSS 17.0 software (Statistical Product and Service Solutions, Chicago, IL, USA) for analysis of variance and Duncan’s test. *p* < 0.05 was used to assign statistical significance. Data are represented as mean values ± standard deviation. Gut microbiota data were analyzed using the free online Majorbio Cloud Platform -. (www.i-sanger.com; accessed on 1 September 2020).

## 3. Results

### 3.1. Effects of Three Cloudy Apple Juices on Weight Lipid Levels and Oxidative Damage

At baseline, the five groups of mice did not differ significantly (*p* > 0.05) in body weight (Table 1). There were no significant differences in food intake and water intake among the five groups, indicating that the three apple juices had no negative effect on the appetite of mice. After 8 weeks of cloudy apple juice consumption, there were significant differences (*p* < 0.01) in body weight (Figure 1A), weight gain (Figure 1B), epididymal fat, and liver index between the HFD group and other groups, indicating that an obesity model induced by a high-fat diet was successfully established and that these three apple juices all had negative effects on the weight gain of mice, while also reducing the accumulation of fat in the liver. The effects of the CAJP group were better than the CAJ group, and the FCAJP showed the greatest effect. The comparison between these three groups showed that polyphenols and *Lactobacillus* are important substances for obesity prevention.

Dyslipidemia refers to increased levels of TC, TG and LDL-C, and a reduced HDL-C level. At week 8, we found that a high-fat diet significantly (*p* < 0.05) increased serum TC, TG, and LDL-C. FCAJP significantly regulated the serum lipid levels by decreasing serum TC (*p* < 0.05), TG (*p* < 0.05), and LDL-C (*p* < 0.05) compared with the HFD group. Meanwhile, the HDL-C content in FCAJP was higher than the other two apple juices (*p* < 0.05). Additionally, no significant differences (*p* > 0.05) were observed in TC content and HDL-C content when supplemented with CAJ and CAJP compared with the HFD group.

GSH-Px and SOD are important free-clearing enzymes in the body, which are positively correlated with free radical scavenging ability in mice [16]. MDA is the end product of lipid peroxidation. As shown in (Table 1), the CAJP and FCAJP groups significantly increased the GSH-Px and SOD, and significantly decreased the MDA compared with the HFD group.

### 3.2. Liver and Epididymal Adipose Tissue Histology Analysis

The average size of EWAT adipocytes in the HFD group was significantly larger than that in the other groups (Figure 2A–E). Additionally, the EWAT adipocytes in the HFD group were incomplete, uneven in shape and size, and arranged irregularly. The cell boundary was clearer, and the size was reduced, with the supplementation of these three apple juices, especially in the FCAJP group.

Histopathological analysis of liver tissue revealed that the cellular structure in the CD group was polygonal, the nucleus was round, the nucleolus was obvious, and the cells were neatly arranged (Figure 2a–e). However, the cellular structure in the HFD group was obviously swollen and arranged in a disorderly manner, the cell outline was blurred. The supplementation of the three apple juices normalized liver damage and well-preserved liver parenchyma compared with the HFD group.

### 3.3. The Richness and Diversity of the Microbial Community

The Shannon diversity index curves were stable for all 40 samples, which suggest that the amount of sequencing data this sequencing is sufficient (Figure 3A). Through the clustering operation, the sequences are divided into many groups according to their similarity, and a group is an operational taxonomic unit (OTU) with 97% similarity levels. Pan and Core were used to describe the changes in the total number of species and core species as the number of samples increased. Eight samples of each group fully covered the species available for analysis, as well as the core species (Figure 3B,C). Different numbers of total OTUs and core OTUs were discovered in the five groups (*n* = 8): 573 total OTUs with 150 core OTUs in the CD group, 536 total OTUs with 53 core OTUs in the HFD group, 577 total OTUs with 57 core OTUs in the HFD + CAJ group, 567 total OTUs with 106 core OTUs in the HFD + CAJP group, and 577 total OTUs with 152 core OTUs in the HFD + FCAJP group, which suggest that a high-fat diet decreased the OTUs and core OTUs, and supplementation of the three kinds of apple juice improved the species richness of the gut microbiota.

The alpha diversity analysis of the sample reflected the richness and diversity of the microbial community [19]. The Sobs (Figure 3D), Ace (Figure 3E), and Chao indexes (Figure 3F) were calculated. The Sobs Chao and Ace indices represent the community richness. As expected, HFD significantly decreased the Sobs (*p* < 0.05) Ace (*p* < 0.05) and Chao index (*p* < 0.05) compared to the CD group. Compared with the HFD group, the HFD + FCAJP group exhibited significantly increased Sobs (*p* < 0.05), Ace and Chao indexes (*p* < 0.01), which means that apple juice with *Lactobacillus* and polyphenols plays a positive role in regulating the richness of the microbial community. *Lactobacillus* and polyphenols significantly increased (*p* < 0.05) the Chao indexes in a comparison of the HFD + CAJ group with the HFD + FCAJP group.

### 3.4. Gut Microbiota Composition at the Phylum and Genus Levels

The gut microbiota composition at the phylum and genus levels in each group is shown in (Figure 4A,B). Firmicutes, Bacteroidota, Desulfobacterota and Actinobacteriota, were the most abundant phylum, and they were present in all groups. Phyllum Firmicutes and phylum Bacteroidetes are the two most important bacterial phyla in the human intestine, accounting for more than 90% [20]. As expected, the HFD group exhibited an increased proportion of Firmicutes (at the phylum level) from 65% to 78% and the Lachnospiraceae_NK4A136_group (at the genus level) from 1.9% to 4.1% alongside a decrease proportion of Bacteroidoata (at the phylum level) from 26% to 11% and norank_f__Muribaculaceae (at the genus level) from 17% to 6.3% compared to the CD group. Mice subjected to three kinds of apple juice treatment showed a difference in gut microbiota profiles, and the ratio of Firmicutes and Bacteroidota decreased compared with HFD groups. The ratio of the Firmicutes and Bacteroidota in the HFD + FCAJP group was the lowest compared to the HFD + CAJ group and the HFD + CAJP group, indicating that *Lactobacillus* and polyphenols can inhibit weight gain by regulating the ratio of Firmicutes and Bacteroidotas.

The greatest proportion of Firmicutes (at the phylum level) was in the HFD group, accounting for 23% while the proportions in the CD, HFD + CAJ, HFD + CAJP and HFD + FCAJP groups were 19%, 22%, 20% and 16% respectively. The greatest proportion of Bacteroidotas (at the phylum level) was in the HFD + FCAJP group, accounting for 30% while the CD, HFD, HFD + CAJ, and HFD + CAJP groups had 28%, 12%, 11%, and 19% respectively. This demonstrated further that a high-fat diet can promote obesity and that *Lactobacillus* and polyphenols can inhibit obesity by regulating the abundance of Firmicutes and Bacteroidotas.

To comprehend more fully the shared richness among each group, a Venn diagram analysis was performed at the genus level (Figure 4C), and the total number of genus in each group was in the range of 162–181. The number of common genus between the HFD + FCAJP and CD samples was 151, between HFD + CAJP and CD samples it was 141, and between HFD + CAJ and CD samples it was 147. There were more genus shared between the HFD + FCAJP and CD samples than between the HFD + CAJ and CD samples, indicating that the bacterial composition of HFD + FCAJP was more similar to that of CD.

Beta diversity analysis was used to explore the similarity or difference of the community composition between different groups by comparing the species diversity of microbial communities in different groups [19]. The principal component analysis (PCA), nonmetric multidimensional scaling (NMDS), principal coordinate analysis (PCoA), and partial least squares discriminant analysis (PLS-DA) results are shown in (Appendix A). A significant difference of microbiota communities was found between the CD group, the HFD group and the HFD + FCAJP group, indicating that the obesity change the gut microbiota composition and FCAJP had a significant impact on the species diversity of microbial communities.

### 3.5. Differences in the Dominant Gut Microbiota

The top 15 species at the phylum level and genus level were selected based on maximum abundance and then a columnar stacked map was drawn, as shown in (Appendix A). It was possible to observe clearly the relative abundance of species and their differences in the five groups. There were no differences between Firmicutes and Bacteroidotas (at the phylum level) between the five groups. The norank_f__Muribaculaceae, Lachnospiraceae_NK4A136_group, Romboutsia, norank_f__Desulfovibrionaceae, and *Lactobacillus* etc. (at the genus level) all exhibited significant differences between the five groups.

The groups were shown in cladograms, and LDA scores of three were confirmed by LEfSe (Appendix A). This tool allows analysis of microbial community data at any clade [21], and statistical analysis was performed from the phylum to the genus level in this study. The HFD + FCAJP group exists in most species and this had a significant influence on the differences between the five groups including Clostridia, Lachnospirales, Lachnospiraceae, Acetatifactor, Lachnospiraceae_UCG_006, Lachnospiraceae_ NK4A136_group, and Lachnoclostridium. For the CD group, Muribaculaceae contributed to the dominant community, and Lactobacillales, Bacilli, and Clostridia contributed to the dominant community and were significantly abundant in the HFD, HFD + CAJ, and HFD + FCAJP groups, respectively. In addition, the results showed that 12, 5, 10, 0, and 29 features with differences were detected in the CD, HFD, HFD + CAJ, HFD + CAJP and HFD + FCAJP groups, respectively. These analyses indicated that FCAJP group enriched more bacterial clades.

### 3.6. Effect of Three Cloudy Apple Juices on SCFAs Concentrations

The effects of three kinds of apple juice on the concentration of SCFAs in the cecal contents were shown in (Table 2). SCFAs are closely related to metabolic functions and contribute to protect the intestinal barrier health; they are produced by intestinal complex carbohydrates fermented by colonic anaerobic microorganisms, and the SCFA content is directly affected by the structure of gut microbiota [22]. In this research, six kinds of SCFA were detected. The results showed that a high-fat diet significantly decreased the levels of six kinds of SCFA compared to the CD group. The administration of three kinds of apple juice, and especially FCAJP significantly increased the levels of six kinds of SCFA. In addition, there was a significant difference between the FCAJP, CAJP and CAJ groups in the levels of propionic acid, isobutyric acid and *n*-valeric acid.

### 3.7. Effects of Three Cloudy Apple Juices on Intestinal Permeability, Intestinal Inflammation and Immunological Barrier

Lipopolysaccharide (LPS), and lipopolysaccharide binding protein (LBP) in serum, reflect changes in the permeability of the intestinal mucosa, which in turn reflect the level of microecological disorders [1]. Secretory immunoglobulin A (sIgA) and fecal calprotectin (FC) in fecal matter were detected to reflect intestinal inflammation. A significant increase in LPS, LBP, sIgA and FC in the HFD group demonstrated that the intestinal permeability was damaged and intestinal inflammation occurred due to the high fat diet. The supplementation of three kinds of apple juice can effectively protect intestinal tract health by decreasing the level of LPS, LBP, sIgA and FC.

To identify the protective effects of three kinds of apple juice, histopathological analysis of the colon was conducted by H&E staining (Figure 5a–e). In the CD group, the colonic villi were arranged neatly and the crypt structure and colonic mucosa structure were relatively complete (Figure 5a). In the HFD group, it was found that the villi of the colon were broken and arranged in a disorderly manner, the crypt structure was damaged, the epithelial mucosa of the colon was edema, and ulceration and shedding occurred in some parts. In addition, the number of goblet cells was significantly reduced, there was obvious inflammatory infiltration, and the villi and crypts of the colon were arranged irregularly, with obvious damage (Figure 5b). The colonic epithelial mucosa structure was slightly ulcerated, but the colonic villi and crypts were arranged in an orderly manner, with no obvious difference in inflammatory infiltration between the HFD + CAJ, HFD + CAJP and HFD + FCAJP groups (Figure 5c–e). The results showed that the administration of three kinds of apple juice can improve colon morphology damage.

## 4. Discussion

The present study has demonstrated that the three apple juices, and particularly the FCAJP, can inhibit the weight gain of mice, reduce fat accumulation, and regulate the blood lipid levels of high-fat diet-induced obese mice by decreasing the ratio of Firmicutes/Bacteroidotas, improving the community richness of the gut microbiota and protecting intestinal tract health. In recent years, many studies have shown that the gut microbiota, which plays an important role in the human system, participate in nutrient absorption and energy regulation. In this context, the mechanism by which three kinds of apple juice can prevent obesity was explored by evaluating the gut microbiota of mice in five groups.

Obesity is characterized by fat accumulation and higher blood lipid levels including TC, TG and LDL-C. TC and LDL-C are considered to be triggers for the development of hyperlipidemia and are closely related to diabetes and metabolic syndrome. In this study, the weight gain, the weight and size of epididymal fat, the liver index, and the TC, TG and LDL-C content all increased significantly in the HFD group (Figure 1, Table 1). With interventions based on three kinds of apple juice, the mice had lower weight gain, liver index, and TC, TG and LDL-C, less epididymal fat and less fat in their liver (Figure 2). Supplementation with FCAJP decreased these parameters more than supplementation with CAJ and CAJP, indicating that FCAJP intervention was more helpful in preventing obesity. The same results were observed by Zhong et al. [22], who reported that blueberry juice fermented by probiotics inhibits obesity and hyperglycemia in high fat diet-fed mice. Park reported similar findings wherein cabbage-apple juice-especially fermented cabbage-apple juice-might have beneficial effects on lipid metabolism dysfunction and obesity-related abnormalities [23]. Such improved properties may be attributed to the rich phenolic acids, viable *Lactobacillus* and other bioactive components such as organic acids in FAJP after fermentation, which were proven to prevent obesity and metabolic disease, regulate gut microbiota, and suppress inflammation [1,10,24]. According to Li [13], *Lactobacillus* fermentation dramatically increased the total phenolic content in juice due to the hydrolytic enzymes of *Lactobacillus* hydrolyzing the complex phenolic compounds into simpler forms.

Considering that obesity induced by a high fat diet is associated with gut microbiota dysbiosis, previous studies have reported that polyphenols and *Lactobacillus* can prevent obesity by regulating the richness and composition of gut microbiota. In this study, a high fat diet significantly decreased the α-diversity including the Chao index, Ace index and Shannon index, which represent the community richness and community diversity, respectively, compared to the CD group (Figure 3D–F). FCAJP significantly increased the α-diversity indicating that it exhibited positive effects on the gut microbiota community richness and community diversity. Many studies have revealed that imbalance of Firmicutes and Bacteroidetes leads to obesity and obesity-related inflammation [24,25]. In this case, the supplementation of CAJ, CAJP and FCAJP decreased the ratio of Firmicutes and Bacteroidetes, while FCAJP had the lowest ratio of Firmicutes and Bacteroidetes (at the phylum level) (Figure 4A). At the genus level, Akkermansia was negatively correlated with inflammation and obesity by reducing lipid accumulation and the concentration of LPS according to the clinical studies [22,26]. As we can see in our study, Akkermansia only occurs in the FCAJP group. This may be because FCAJP is rich in polyphenols and *Lactobacillus* and was therefore capable of enriching the Akkermansia population [27]. Furthermore, most of the Bacteroides in the intestine regulate the energy metabolism of the host [25]. The relative abundance of Bacteroides in the five groups was 3.84%, 2.44%, 2.04%, 5.88%, and 7.79% respectively, indicating that HFD decreased the relative abundance of Bacteroides, and CAJP and FCAJP supplementation augmented the relative abundance of Bacteroides, which is consistent with Lau, E [28]. The increase of *Lactobacillus* in the HFD group and the decrease in the CAJ, CAJP and FCAJP groups was not consistent with previous research [1,22]. Although most *Lactobacillus* showed anti-obesity activity, *Lactobacillus* has many subspecies, some of which are related to weight gain and can promote obesity due through inducing a deficiency in some important enzymes [25].

The richest SCFAs in the cecum were acetic, butyric and propionic acid. Butyric acid was the main energy source of colon cells, which could cause fatty acid oxidation, lipolysis and thermogenesis and improve insulin sensitivity. Propionic acid and acetic acid were transported to the liver by the portal vein. In addition, propionate induced and participated in gluconeogenesis in the liver, while acetate restricted fat accumulation and increased the expression of genes related to thermogenesis [24]. In the study, a high-fat diet for 8 weeks significantly reduced the content of acetic, propionic isobutyric, butyric, isovaleric, and n-valeric acid. Moreover, CAJ, CAJP and FCAJP notably upregulated them, and FCAJP increased the content of SCFAs more than CAJ and CAJP, indicating that FCAJP showed the most potential in regulating gut microbiota (Table 2). Furthermore, the increase of SCFAs due to three kinds of apple juice could restrict fat accumulation and regulate lipid levels, as reflected in the decrease in weight gain and the decrease in the level of TC, TG and LDL-C. The spearman’s correlation analysis between the gut microbiota, SCFAs and other related indicators also validates this view. Gut microbiota that were positively related to blood lipids were negatively related to SCFAs (Appendix A). In addition, intestinal barrier dysfunction would directly cause an increase in plasma LPS and LBP concentration. In this research, the HFD group had considerably increased amounts of LPS and LBP and manifested deterioration in the health of the intestinal tract through histopathological observations of colon sections (Figure 5) and inflammation indices including FC and sIgA. Supplementation with the three kinds of apple juice protected the intestinal tract health via reducing the concentrations of LPS and increasing SCFAs. The cause of these phenomena may be that the SCFA-producing bacteria Bacteroides, Allobaculum, and norank_f__Muribaculaceae were upregulated and the LPS-producing bacteria Turicibacter and Lachnoclostridium were downregulated [24]. Bacteroides is positively related to butyric acid (r = 0.38, *p* < 0.05) and isovaleric acid (r = 0.34, *p* < 0.05) (Appendix A). PICRUSt was used to predict the functionality of a metagenome (Appendix A). There were 24 functionalities analyzed in 40 samples. Amino acid transport and metabolism, carbohydrate transport and metabolism, and general function prediction were the three most enriched functions. HFD decreased all functionalities including coenzyme transport and metabolism, lipid transport and metabolism, cell wall/membrane/envelope biogenesis, posttranslational modification, protein turnover, chaperones, inorganic ion transport and metabolism, secondary metabolite biosynthesis, transport and catabolism, intracellular trafficking, secretion, and vesicular transport, while FCAJ increased them. In conclusion, FCAJ has the greatest potential to prevent obesity via modulating gut microbiota and protecting intestinal tract health and should be developed further as a functional apple juice. In addition, the addition of polyphenols and fermentation by *Lactobacillus* can increase the effectiveness of the apple juice against obesity.

## 5. Conclusions

The study showed that a high-fat diet could lead to the accumulation of epididymal white adipose, disturbance of gut microbiota and hyperlipidemia such as increased levels of TC, TG, and LDL-C. However, the three kinds of apple juice, especially the FCAJP, inhibited weight gain, regulated blood lipid levels, maintained the balance of gut microbiota and protected intestinal tract health effectively. In conclusion, FCAJP can be used as an effective green intervention in preventing obesity.

## Figures and Tables

**Figure 1 nutrients-13-00971-f001:**
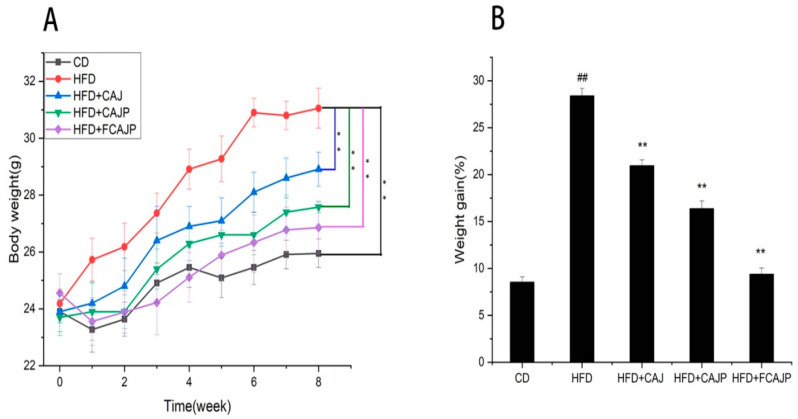
(**A**) Body weight of five groups in different periods of 8 weeks and (**B**) weight gain of five groups (*n* = 8). Note: CD: basal diet & Saline; HFD: 45% high-fat diet & Saline; HFD + CAJ: 45% high-fat diet & CAJ; HFD + CAJP: 45% high-fat diet & CAJP; HFD + FCAJP: 45% high-fat diet & FCAJP, ** *p* < 0.01 versus HFD; ## *p* < 0.01 versus CD.

**Figure 2 nutrients-13-00971-f002:**
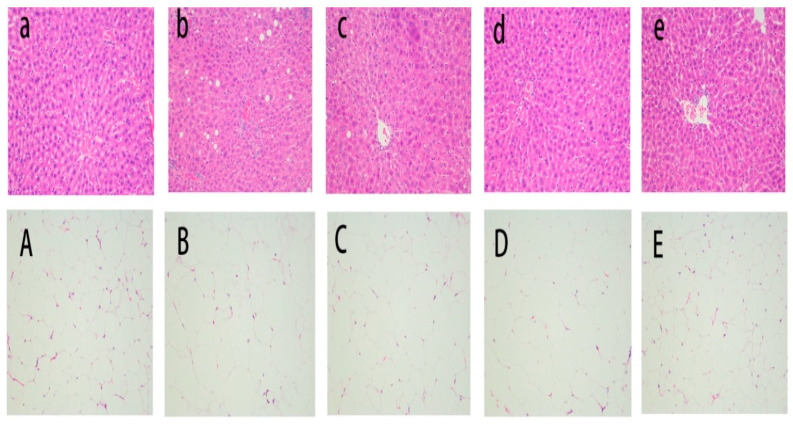
Effects of three cloudy apple juices on high-fat diet-induced histopathological alterations in the liver (**a**–**e**) and epididymal white adipose tissue (**A**–**E**) structure after H & E staining (200×). Note: (**A**,**a**): basal diet & Saline; (**B**,**b**): 45% high-fat diet & Saline; (**C**,**c**): 45% high-fat diet &CAJ; (**D**,**d**): 45% high-fat diet & CAJP; (**E**,**e**): 45% high-fat diet & FCAJP.

**Figure 3 nutrients-13-00971-f003:**
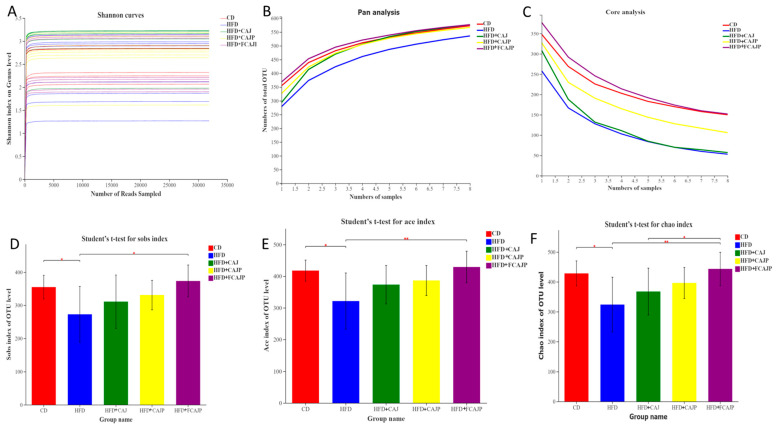
Responses of the diversity, richness, and structure of the gut microbiota (*n* = 8). (**A**) Shannon curves; (**B**) Pan analysis; (**C**) Core analysis; (**D**) Comparisons between Sobs indexes of five groups; (**E**) Comparisons between Ace indexes of five groups; (**F**) Comparisons between Chao indexes of five groups. Note: CD: basal diet & Saline; HFD: 45% high-fat diet & Saline; HFD + CAJ: 45% high-fat diet & CAJ; HFD + CAJP: 45% high-fat diet & CAJP; HFD + FCAJP: 45% high-fat diet &FCAJP. Differences were assessed by student’t *t*-test and denoted as follows: * *p* < 0.05 and ** *p* < 0.01 in a comparison between the two groups. Values are presented as mean ± SEM.

**Figure 4 nutrients-13-00971-f004:**
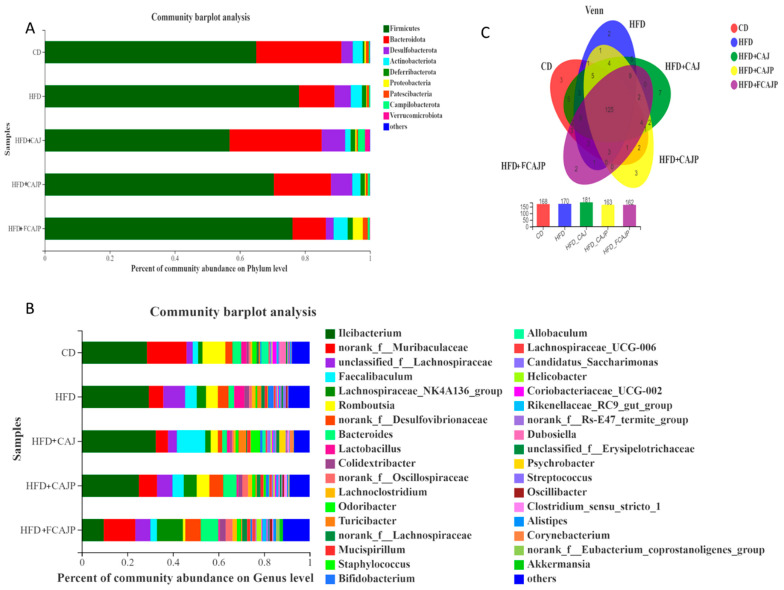
Responses of the diversity, richness, and structure of the gut microbiota (*n* = 8). (**A**,**B**) Relative abundances of the gut microbiota at the phylum level and genus level; (**C**) Venn diagram analysis at genus level. Note: CD: basal diet & Saline; HFD: 45% high-fat diet & Saline; HFD + CAJ: 45% high-fat diet & CAJ; HFD + CAJP: 45% high-fat diet & CAJP; HFD + FCAJP: 45% high-fat diet & FCAJP.

**Figure 5 nutrients-13-00971-f005:**
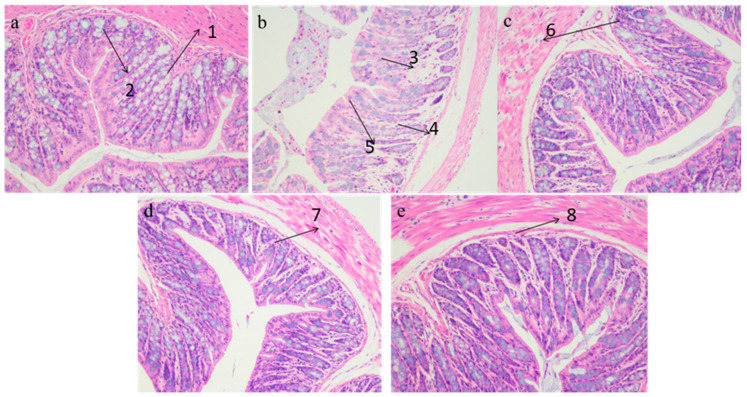
Effects of three cloudy apple juices on high-fat diet-induced histopathological alterations in the colon (200×). Note: (**a**): basal diet & Saline; (**b**): 45% high-fat diet & Saline; (**c**): 45% high-fat diet & CAJ; (**d**): 45% high-fat diet & CAJP; (**e**): 45% high-fat diet &FCAJP. Arrow 1: colonic mucosa structure were relatively complete; Arrow 2: the crypt structure were relatively complete; Arrow 3: the villi of the colon were broken and arranged in a disorderly manner, the crypt structure was damaged; Arrow 4: the epithelial mucosa of the colon was edema, and ulceration and shedding occurred in some parts; Arrow 5: villi and crypts of the colon were arranged irregularly, with obvious damage; Arrow 6–8: The colonic epithelial mucosa structure was slightly ulcerated, but the colonic villi and crypts were arranged in an orderly manner, with no obvious difference in inflammatory infiltration.

**Table 1 nutrients-13-00971-t001:** Influences of three cloudy apple juices on growth performance, serum and hepatic parameter.

	CD	HFD	HFD + CAJ	HFD + CAJP	HFD + FCAJP
Food intake (g/day)	2.09 ± 0.20 ^a^	2.15 ± 0.15 ^a^	2.01 ± 0.23 ^a^	1.99 ± 0.14 ^a^	2.05 ± 0.16 ^a^
Water intake (mL)	3.25 ± 0.31 ^a^	3.21 ± 0.32 ^a^	3.43 ± 0.22 ^a^	3.39 ± 0.41 ^a^	3.22 ± 0.31 ^a^
Epididymal fat (g)	0.753 ± 0.13 ^a^	2.01 ± 0.94 ^b^	1.44 ± 0.16 ^c^	1.304 ± 0.22 ^c^	0.84 ± 0.56 ^a^
Liver index (%)	3.24 ± 0.34 ^a^	3.76 ± 0.18 ^b^	3.35 ± 0.18 ^a^	3.22 ± 0.28 ^a^	3.18 ± 0.32 ^a^
TC (mmol/L)	3.36 ± 0.39 ^a^	4.73 ± 0.33 ^b^	4.50 ± 0.61 ^b^	4.42 ± 1.00 ^b^	3.45 ± 0.37 ^a^
TG (mmol/L)	0.52 ± 0.03 ^a^	0.75 ± 0.07 ^c^	0.58 ± 0.08 ^ab^	0.55 ± 0.05 ^ab^	0.59 ± 0.07 ^ab^
HDL-C(mmol/L)	2.82 ± 0.23 ^b^	2.54 ± 0.20 ^a^	2.57 ± 0.28 ^a^	2.51 ± 0.44 ^a^	2.72 ± 0.14 ^b^
LDL-C (mmol/L)	0.33 ± 0.04 ^b^	0.52 ± 0.04 ^a^	0.45 ± 0.03 ^c^	0.44 ± 0.05 ^c^	0.35 ± 0.06 ^b^
Liver GSH-Px	555.68 ± 18.54 ^a^	548.31 ± 32.32 ^a^	629.48 ± 28.12 ^b^	690.83 ± 45.74 ^c^	612.37 ± 28.25 ^b^
Liver SOD (U/mgprot)	1100.96 ± 61.81 ^a^	1027.59 ± 69.32 ^a^	1093.68 ± 108.28 ^a^	1358.18 ± 67.77 ^b^	1303.89 ± 57.71 ^b^
Liver MDA (nmol/mgprot)	0.77 ± 0.06 ^a^	0.85 ± 0.22 ^b^	0.75 ± 0.06 ^a^	0.49 ± 0.11 ^c^	0.51 ± 0.06 ^c^

Note: Differences between groups were analyzed by Duncan’s test. In each column mean ± SD values (*n* = 8) bearing different letters differ significantly (*p* < 0.05). Note: CD: basal diet & Saline; HFD: 45% high-fat diet &Saline; HFD + CAJ: 45% high-fat diet & CAJ; HFD + CAJP: 45% high-fat diet & CAJP; HFD + FCAJP: 45% high-fat diet & FCAJP. TC: Total cholesterol; TG: triglyceride; HDL-C: high-density lipoprotein cholesterol; LDL-C: low-density lipoprotein cholesterol; GSH-Px: glutathione peroxidase; SOD: superoxide dismutase; MDA: malondialdehyde.

**Table 2 nutrients-13-00971-t002:** Effect of three cloudy apple juices on SCFAs concentrations (μg/g), intestinal permeability and intestinal inflammation.

	CD	HFD	HFD + CAJ	HFD + CAJP	HFD + FCAJP
Acetic acid	687.74 ± 100.33 ^a^	489.94 ± 89.35 ^b^	550.39 ± 100.27 ^c^	570.39 ± 90.29 ^c^	620.67 ± 121,94 ^ac^
Propionic acid	101.02 ± 51.24 ^a^	70.44 ± 21.46 ^b^	79.33 ± 16.76 ^b^	77.98 ± 10.57 ^b^	87.35 ± 16.44 ^c^
Isobutyric acid	76.64 ± 16.54 ^a^	52.68 ± 14.41 ^b^	60.49 ± 13.11 ^c^	68.44 ± 10.47 ^ac^	74.98 ± 12.21 ^a^
Butyric acid	87.21 ± 13.72 ^a^	73.51 ± 19.22 ^b^	80.35 ± 11.53 ^ab^	82.42 ± 15.14 ^a^	90.21 ± 19.72 ^a^
Isovaleric acid	71.59 ± 11.65 ^a^	51.38 ± 10.45 ^b^	62.63 ± 10.15 ^c^	60.59 ± 15.65 ^c^	66.89 ± 11.25 ^ac^
N-valeric acid	114.14 ± 16.52 ^a^	94.79 ± 23.52 ^b^	99.14 ± 12.58 ^b^	107.34 ± 13.82 ^ac^	116.14 ± 20.12 ^ac^
LBP(ng/mL)	10.36 ± 1.53 ^a^	16.05 ± 1.93 ^b^	11.68 ± 1.75 ^c^	11.41 ± 1.39 ^c^	10.99 ± 0.86 ^ac^
LPS(ng/mL)	160.69 ± 15.93 ^a^	212.97 ± 20.11 ^b^	195.86 ± 21.11 ^c^	200.52 ± 17.43 ^c^	189.31 ± 10.44 ^c^
FC (pg/g)	51.01 ± 7.87 ^a^	57.6 ± 13.06 ^b^	54.25 ± 9.65 ^ab^	49.77 ± 13.35 ^ac^	44.77 ± 9.93 ^d^
sIgA(μg/g)	0.92 ± 0.21 ^a^	0.97 ± 0.38 ^b^	0.62 ± 0.31 ^c^	0.81 ± 0.42 ^d^	0.51 ± 0.27 ^e^

Note: Differences between groups were analyzed by Duncan’s test. In each column mean ± SD values (*n* = 8) bearing different letters differ significantly (*p* < 0.05). Note: CD: basal diet & Saline; HFD: 45% high-fat diet & Saline; HFD + CAJ: 45% high-fat diet & CAJ; HFD + CAJP: 45% high-fat diet & CAJP; HFD + FCAJP: 45% high-fat diet & FCAJP. LBP: lipopolysaccharide binding protein; LPS: lipopolysaccharide; sIgA: secretory immunoglobulin A; FC: fecal calprotectin.

## Data Availability

The study didn’t report any data.

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
