# Peer review of "Cloudy Apple Juice Fermented by Lactobacillus Prevents Obesity via Modulating Gut Microbiota and Protecting Intestinal Tract Health"

_nutrients, 2021, doi:10.3390/nu13030971_

Round 1

Reviewer 1 Report

This study demonstrated positive effects of three different cloudy apple juice (CAJ) compositions in the prevention of body and fat weight gain and in the improvement of lipid profile and liver steatosis in mice fed with high-fat diet. Of interest, these positive effects were associated to reduced intestinal dysbiosis, permeability and inflammation and increased production of short chain fatty acids. However, if some intestinal effects of CAJ compositions might explain the mechanism of their anti-obesity effect remains to be elucidated.

1- Additional analysis should be performed to understand how the effects of three apple juice tested on gut microbiota composition, short fatty acid composition and intestinal permeability and inflammation could contribute to HFD-induced weight gain prevention and to serum lipid profile and liver steatosis improvement. These might include:

  • To analyse markers of adipogenesis (Adipoq, Plin1, Fabp4, Slc2a4, Pparg), inflammation (Il6, Tnf, Mcp1) and thermogenesis (Ucp1, Pgc1a, Dio2, Prdm16) in fat depots (including inguinal and visceral (mesenteric or perigonadal) white adipose tissue).
  • To analyse lipogenesis-, fatty oxidation-, gluconeogenesis- and mitochondrial biogenesis- related gene expression in liver samples.
  • Then data from these gene expression analyses should be correlated with microbiota abundance, short chain fatty acids levels, and LBP, LPS, FC and sIgA levels to elucidate which CAJ-induced change or changes could underlie the anti-obesogenic effect of these nutritional interventions.
  • 2- In my view, serum glucose levels also should be measured to explore the possible effect of these interventions on hyperglycemia.
  • 3- Plasma LBP levels are detected around 3-15 ug/ml in normal conditions, but in table 2 the units are ng/ml. These data should be revised.
  • 4-Whereas LBP levels were normalized with three apple juice, LPS levels in these treatments were more similar to HFD group compared to CD. Could the authors explain this discrepancy?
  • 5- The quality of Figure 3 and 4 should be improved. The font on the labels of the graphs or text is too small and cannot be read well.

Author Response

Dear Reviewer,

We would like to express our sincere appreciation for effort you have spent on our manuscript entitled Cloudy apple juice fermented by lactobacillus prevents obesity via modulating gut microbiota and protecting intestinal tract health” (No. nutrients-1135915). Thank you very much for the positive and constructive comments and suggestions, which are very helpful for the improvement of our manuscript. We have considered the comments carefully and have revised the manuscript thoroughly accordingly. We hope that the revisions and responses will meet with the comments. The point to point responses to the comments are listed after this letter. We look forward to your first reply.

Best regards,

Yours sincerely,

Zhenpeng Gao

Reviewer 2 Report

Comments to the Authors of manuscript number: nutrients-1135915  entitled “Cloudy apple juice fermented by lactobacillus prevents obesity via modulating gut microbiota and repairing intestinal mucosal integrity”.

  1. L 40 what is a microbiota structure?
  2. L 42 health of microbiota?
  3. L 69 – the study
  4. L 82- 84 – how the content of these acids were determined? It should be explained
  5. L 85 – how it was extracted? It should be explained
  6. L 88 -91 – how the content of these acids were determined? It should be explained
  7. L 74 – city lacks for Granny Smith Apple and Qin Guan
  8. L 74 – “Borkh. )”
  9. L 95- 97 how the content of these acids were determined? It should be explained
  10. The number of ethical permission should be given. Please indicate by whom the permit was issued
  11. L 103 – top water?
  12. L 94 – how many CFU was used for each bacteria?
  13. L 107 – how CFU was calculated?
  14. L 92 – was the fermentation stopped before administration to animals?
  15. L 108 – how this amount was determined? It should be explained
  16. L 115 – distilled water? How mice can drink distilled water?! potential health consequences may arise. Some problems occur because of the lack of minerals and potential changes to the body’s balance of electrolytes, fluid, minerals and pH.
  17. “blood samples were collected from the orbit”
  18. L 134 – what is EWAT
  19. results it is a part, where only data are presented without discussion with other data from the literature
  20. Table 1. What is liver index? It should be explained
  21. L 215 – in HE staining there is not possible to determine fat droples in the liver
  22. L 217 - in HE staining there is not possible to distinguish lipid vacuoles
  23. L 216 – based on this result it is not possible to conclude in this manner
  24. L 228 – “OUT” should be explained
  25. part of 3.3 – the analysis of gut microbiota should be described in detiles in the material and methods. There is some information which are unclear
  26. L 365 – how this colon damage was determined? It should be explained. There are not presented histomorphological parameters.
  27. Figure 5 what is indicated by black arrows?
  28. L 375 there was not determined repairing intestinal mucosa integrity
  29. There is a lack of data if apple juice of each kind was freshly prepared before administration to mice. Or it will collected. Thus, how the time influenced the content of the juice?

Author Response

Dear Reviewer,

We would like to express our sincere appreciation for effort you have spent on our manuscript entitled Cloudy apple juice fermented by lactobacillus prevents obesity via modulating gut microbiota and protecting intestinal tract health” (No. nutrients-1135915).Thank you very much for the positive and constructive comments and suggestions, which are very helpful for the improvement of our manuscript. We have considered the comments carefully and have revised the manuscript thoroughly accordingly. We hope that the revisions and responses will meet with the comments. The point to point responses to the comments are listed after this letter. We look forward to your first reply.

Best regards,

Yours sincerely,

Zhenpeng Gao

Round 2

Reviewer 2 Report

I have no additional comments.